# Community health worker and caregiver experiences and perceptions of a multimodal handheld pulse oximeter used in sick child consultations in rural Burundi: A qualitative evaluation

Sarah Bauler [1,2]*, Leocadie Nivyindika[3,4], Titus Kirwa[1], Vital Habonimana[3], Dionis Nizigiyimana[5], Miles A. Kirby[6,7], Asrat Tolossa[8]

**1** World Vision International, London, England, **2** Department of International Health, Johns Hopkins University, Baltimore, Maryland, United States of America, **3** World Vision Burundi, Bujumbura, Burundi, **4** United Nations International Children's Emergency Fund, Bujumbura, Burundi, **5** National Institute of Public Health, Bujumbura, Burundi, **6** World Vision United States, Washington, DC, United States of America, **7** Department of Global Health and Population, Harvard University, Boston, Massachusetts, United States of America, **8** World Vision Canada, Mississauga, Canada

* sarah_bauler@wvi.org

## Abstract

Community Health Workers (CHWs) in low- and middle-income countries are essential in providing primary health care to remote communities. However, due to limited diagnostic tools, CHWs often struggle to correctly identify childhood illnesses, especially pneumonia. We conducted a prospective pilot study and used qualitative research methods to evaluate acceptability and feasibility of a multimodal pulse oximeter used by CHWs during their integrated community case management (iCCM) of childhood illness consultations in rural Burundi. We used purposive sampling to recruit CHWs and trained them to use the oximeters during household iCCM consultations for children 6–59 months of age. After eight weeks of using the devices, we conducted eight focus group discussions to evaluate experiences and perceptions of the device among CHWs and caregivers. Our thematic analysis, based upon deductive and inductive reasoning, identified the following themes: durability, storability, trust, self-efficacy, child agitation, ease of using the device, and interpretation of parameters. CHWs deemed the devices highly acceptable and took pride in safely storing them but reported challenges in utilizing respiration rate, pulse, and oxygen saturation (though temperature was understood). Child agitation was a barrier to oximeter use, especially among children 6–12 months. Additional CHW capacity-building on interpreting parameters is needed when using oximeters during household iCCM consultations in Burundi, including an iCCM job aid (decision-making tree) with oxygen saturation and respiratory rate cut-offs for treatment and/or referral. Training and using child-calming techniques could be an important strategy for obtaining quality measurements. While CHWs and caregivers highly valued the oximeters during sick child visits, the devices may be better utilized and scalable at the health facility level.

**Data availability statement:** The data supporting this study are not publicly available due to ethical restrictions and institutional regulations. Interested researchers may request access to the data by contacting the Burundi National Institute of Public Health, Avenue de l'Hôpital no 3 I B.P. Bujumbura, Burundi. Requests will be reviewed by the Burundi Bioethics Committee for Health to ensure compliance with ethical guidelines and data protection regulations before any data sharing can occur.

**Funding:** This study was funded by the European Union (EDF/2019/405-314). The funders had no role in the study design, data collection and analysis, decision to publish, or preparation of the manuscript.

**Competing interests:** The authors have declared that no competing interests exist.

## 1. Introduction

Globally, the leading causes of mortality among children younger than five years of age are pneumonia (15%), diarrhea (8%), malaria (5%), and newborn sepsis (7%) [1]. In Burundi, the fifth lowest country on the Human Development Index [2], malaria, lung disease (including pneumonia), and acute diarrhea account for an estimated 93% of all hospitalizations of children ages 1 to 59 months [3]. Symptoms of malaria, diarrhea, and pneumonia can manifest in similar ways. A fever is a common symptom of both malaria and pneumonia, and diarrhea can occur among children suffering from malaria and pneumonia. Early identification, rapid referral, and treatment of sick children when appropriate are foundational to integrated community case management (iCCM), especially in Burundi, where the risk of dying before age five is almost 1.6 times higher than in the WHO African region [3].

Challenges to correctly identifying childhood illnesses, especially among Community Health Workers (CHWs), can lead to over-prescribing antibiotics, especially amoxicillin. A cross-sectional study of sick children younger than five years who attended health facilities in eight low-to-middle-income countries (LMICs)—Haiti, Kenya, Malawi, Namibia, Nepal, Senegal, Tanzania, and Uganda—found children received, on average, 25 antibiotic prescriptions during their first five years of life [4]. An excessive amount could harm children's ability to fight pathogens and increase antibiotic resistance worldwide [4]. Antimicrobial resistance (AMR) can destroy the natural gut flora, weaken a child's ability to fight harmful pathogens, and increase the risk of disease spread, severe illness, and mortality [5]. In 2019, the World Health Organization (WHO) declared AMR one of the top ten public health threats threatening humanity [6]. One approach to reducing risk of AMR is to improve antibiotic prescribing practices through more accurate diagnosing.

While iCCM has been widely adopted throughout LMICs in Asia and Africa as a strategy to identify and treat childhood illnesses, a recent meta-analysis of iCCM studies found that iCCM had little to no effect on neonatal infant mortality and the effect on infant and under-five mortality is uncertain [1]. A significant barrier to quality iCCM services is CHWs often lack effective tools to diagnose childhood illnesses correctly, especially pneumonia [7]. Accurate and reliable counting of respiration rate (RR), a biomarker for pneumonia, in young children is especially difficult due to human error [8]. CHWs in LMICs diagnose pneumonia among children based on increased RR or difficulty breathing [9]. However, studies have found that CHWs counting RR in Uganda only correctly diagnose or treat 40% of all cases of childhood pneumonia [8]. Furthermore, a multicenter evaluation of RR counting aids in Cambodia, Ethiopia, South Sudan, and Uganda using the Mark TWO ARI timer, counting beads with an ARI timer, and a mobile phone Respirometer found that none of the devices performed well based on agreement with the reference standard [9]. Agreement with the reference standard was only 8 to 20% among young infants [8], which is problematic as children under two years are most at risk for poor pneumonia outcomes [10].

The WHO and UNICEF iCCM guidelines provide a diagnostic roadmap [11] for CHWs and frontline health workers to assess children with respiratory distress by counting the child's breaths per minute (over 50 breaths for children younger than one year and over 40 breaths for older children). However, CHWs often struggle to count the respiratory rate of a distressed child, especially if that child is upset and crying. Another sign of pneumonia is cyanosis (e.g., skin turning blue due to a shortage of oxygen), which is often more difficult to detect in children with darker-colored skin [12]. Also, severely anemic infants may not appear cyanotic until their oxygen saturation is critically low [13]. Community Health Workers rarely have a way to test the oxygen saturation levels in the blood; thus, many children may not be appropriately referred and may not receive the oxygen treatment they urgently need. Nurses in Burundi are only equipped with one-minute stopwatches (Mark Two ARI Timers) and

mercury thermometers at the health clinic level, which impedes their ability to triage and identify children needing urgent health services. Pediatric pulse oximeters to identify hypoxemia are rarely available in LMICs, a barrier to promptly detecting and treating children with pneumonia [14]. Hypoxemia with pneumonia is also associated with an increased risk of mortality [15] and is essential for triaging children (and adults), especially in fragile contexts where oxygen supplies are scarce [16]. Use of pulse oximetry alongside Integrated Management of Childhood Illness (IMCI) can also reduce antibiotic prescription rates in clinical settings, as found in a recent study in Malawi [17]. Though use of pulse oximetry and automatic respiratory rate counters is slowly increasing in health facilities in LMICs [18,19], there is potential for the use of these tools among frontline CHWs to facilitate accurate diagnoses and timely referrals. However, there is a lack of evidence on the feasibility, perceptions, and acceptability of this technology among CHWs and child caregivers, particularly in underserved areas such as rural Burundi.

## 2. Materials and methods

### 2.1. Research objectives

To strengthen iCCM and early identification and referral of sick children in low-resource settings, World Vision Burundi, in conjunction with the Ministry of Health (MOH), conducted a prospective pilot study and used qualitative research to explore the acceptability and feasibility of a non-invasive pulse oximeter used during CHW iCCM consultations. Our primary research question was, "What are the barriers (obstacles) and facilitators (opportunities) to utilizing pulse oximetry during sick child visits conducted by community health workers in low-resource settings in Burundi? To help answer our research question, we selected a rugged, portable, handheld device called the Rad-G Pulse Oximeter (Fig 1), which can measure pulse rate (PR), oxygen saturation (SpO$_2$), plethysmography-based respiration rate (RRp), perfusion index (PI), and temperature (temp) for our study [20]. The device includes a clinical-grade infrared thermometer to measure temperature, an external casing built to withstand 6 feet drops, 24 hours of rechargeable battery life, a lightweight and slim profile for easy transport and storage, and a paediatric probe (Fig 1). A prospective, multi-centre, single-blinded, trial in Cambodia, Ethiopia, South Sudan, and Uganda found that the Masimo oximeter (one phone) performed best compared the Contec and Devon oximeters (two finger-tip) and Lifebox and Utech oximeters (handheld) among frontline health workers in low resource settings [21]. A study at a tertiary hospital in India found a high degree of agreement between plethysmography RR using the Rad-G device and physician-measured RR, indicating the oximeter provided reliable and accurate measurements [22].

### 2.2. Study setting

The study was conducted in Burundi, a mountainous, land-locked country in East Central Africa, bordering the Democratic Republic of the Congo, Tanzania, and Rwanda. To our best knowledge, this is the first study to evaluate opportunities and barriers to using pulse oximetry among frontline health workers in Burundi.

CHWs in Burundi are volunteers [23] and receive one week of theoretical training and one week of practical training (personal communication, Nivyindika, 2023). One CHW is assigned to one sub-hill (sub-village) [24] and provides community health services to 50 to 200 people on average and are given a wooden or metal trunk to store medications (Briggs, 2014; World Vision International, 2017). Medicines are supplied and restocked from the nearest health facility and include amoxicillin (for treatment of pneumonia), Artemisinin-based combination therapies (for treatment of malaria), oral rehydration solutions (for treatment of

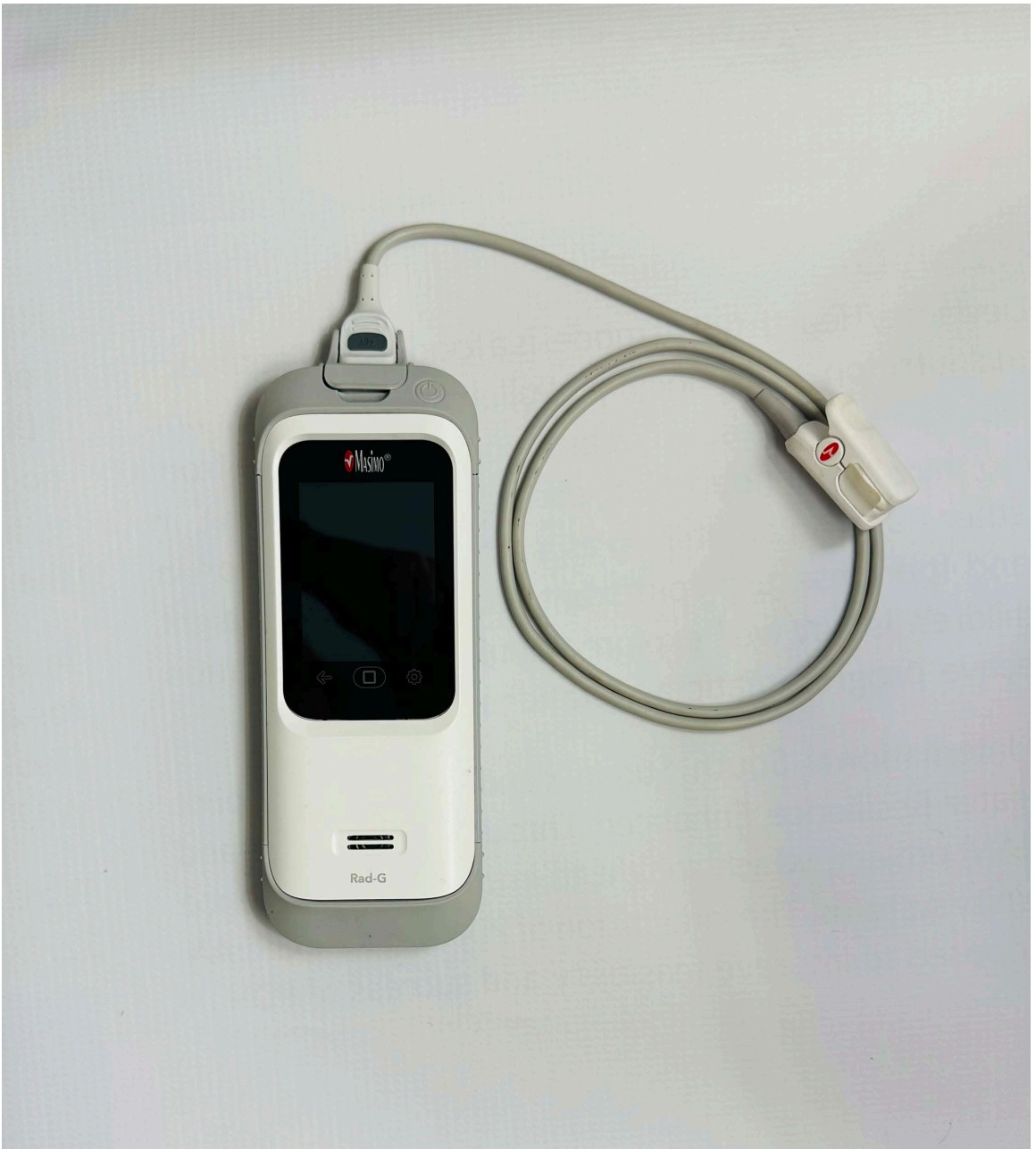

**Fig 1. Masimo's Rad-G pulse oximeter and reusable clip probe (sensor) used by CHWs during their iCCM consultations.**

diarrhea), and albendazole (for treatment of worms) [25]. Due to the rugged and mountainous topography, caregivers often bring their sick children to the CHW's home to alleviate the travel burden upon CHWs. While CHWs record medications they dispensed, they lack sick child recording forms (personal communication, Nivyindika, 2023). On average, CHWs conduct five iCCM consultations per day, or 20 to 25 consultations per week. CHWs may overprescribe medications when unsure of a child's illness diagnosis, treating a child for malaria and pneumonia simultaneously to minimize the risk of the caregiver having to bring back the child (personal communication, Tolossa, 2023).

## 2.3. Sampling and data collection

We used purposive sampling to recruit 32 volunteers, MOH-supported CHWs tasked with providing iCCM services to children 6 to 59 months of age to participate in the study (16 CHWs in Rutana Province and 16 CHWs in Cankuzo Province). The CHWs were recruited on October 31, 2022 and gave written consent to participate in the study and to protect the devices; the study ended March 3, 2023. CHWs were trained on how to use the device during their household iCCM consultations by World Vision Burundi and MOH staff. The two-day training, conducted in Kirundi on November 3 and 4, 2022, covered the device components and features, charging the device, an overview of the displays, sensor application using the finger, performing a screening/spot-check, device maintenance, and troubleshooting. CHWs were also given the opportunity to practice with the device. Initially, we planned to assign one Rad-G device to four CHWs, but the mountainous topography in the provinces and distances between CHWs did not make sharing the devices feasible. CHWs piloted the devices in their assigned sub-hills (villages) for eight weeks and were instructed to use the device for all sick child encounters.

Based upon empirically-based findings and thematic analysis [26], a sample size of four CHW and four caregiver focus group discussions (FGDs), with eight participants per FGD, were conducted on January 24 and 25, 2023 in Rutana and Cankuzo Provinces to sufficiently reach thematic saturation. Verbal consent forms were obtained from CHWs and Caregivers and documented by the FGD Facilitator and witnessed by the FGD Notetaker; both provided their names and signatures to the verbal consent form. Photo consent forms were obtained for all study photos. The CHW and Caregiver FGD guides were developed to explore the experiences, perceptions, and specific attributes of the device (i.e., portability, storability, durability) and recommendations for improving the device during CHW iCCM consultations. The Caregiver FGD guide included questions exploring caregivers' perceptions of how well they and their children liked the device, as well as community perceptions and experiences around sick child encounters. Each FGD included a facilitator trained in group dynamics, an observer, and a notetaker and were conducted and recorded in Kirundi and translated into French and English.

## 2.4. Data analysis

The translated English transcripts were uploaded into the qualitative analysis software MAX-QDA (VERBI Software, Berlin, Germany) as a tool to organize data. We then used deductive and inductive reasoning to develop CHW and caregiver codebooks that included descriptive, process, emotional, value, and concept codes. The codebooks were used to identify participant responses that fit a specific code, and some responses fell under two or more codes. We primarily used horizontal analysis to find emerging themes and sub-themes across all eight transcripts.

## 2.5. Ethical approvals

Ethical clearance for this study was obtained from the Burundi National Ethics Committee for the Protection of Human Beings participating in behavioral research and the statistical visa from the Burundi Institute of Statistics and Economic Studies (ISTEEBU) prior to commencing the project. A pilot committee consisting of nine doctors from the MOH and World Vision Burundi was established to supervise the study and to determine if the study findings justified efforts to scale the innovation and adapt iCCM protocols. A technical committee consisting of delegates from the MOH, UNICEF, WHO, and World Vision Burundi was also established to provide quality assurance to the study

protocol, instruments, and study findings. Finally, photo consent forms were obtained for all photos.

## 3. Results

Thirty CHWs, 15 each from Cankuzo and Rutana Provinces, participated in the FGDs (18 female and 12 male CHWs). They ranged in age from 23 to 60 years (mean age 41.5 years); only two CHWs had attended secondary school. Thirty caregivers, 14 in Cankuzo and 16 in Rutana Provinces, participated in the FGDs (27 female and three male caregivers). The oldest caregiver was 43, while the youngest was 20 (mean age 31.5 years); no caregivers reported completing secondary school.

### 3.1. Caregivers' perceptions of the device

Many CHWs and caregivers noted that children had adverse perceptions of the device before using it, thinking the oximeter was a vaccination or the sensor's infrared light would burn them. One CHW stated:

'The children were scared at first. They thought we were going to stick them with needles (vaccination) like we used to do for the thick drop.'

Another caregiver stated:

'When children first saw these devices, they were afraid. And when I saw it for the first time, inside, there was a light that looked like fire; we thought it could burn the child.'

However, FGD participants noted that most children calmed down once the probe was placed on the child's finger. Caregivers also perceived the device as a new technology that could help frontline health works identify diseases more accurately and quickly. One caregiver stated:

'We have seen that development has embraced us.'

### 3.2. Device durability and storability

Over the eight weeks pilot period, no incidences of the device breaking were reported. However, the CHWs noted that the probes were more difficult to keep safe, as there was no mechanism to attach them to the device safely (they were left dangling). Most CHWs did not need to recharge their oximeters over the 8-week pilot period, with only one CHW in Rutana Province charging the device twice. However, CHWs lacked portable backup battery backs to charge the devices, requiring them to walk several miles to the health facility to charge their devices (charging was free). Several CHWs noted that they did not have a safe place to recharge their devices and feared they could be stolen. To prevent theft, CHWs stored their devices in locked metal or wooden trunks provided by the MOH (Fig 2), keeping them safe from children and theft.

We also explored the ease of keeping the devices clean. Many participants noted that the device was relatively easier to keep clean than the probe, which dirtied quickly. There was no preferred method to clean the device, and participants explained the various techniques they used to clean the device ranging from using a piece of cloth to wiping it with a loincloth. From our observations, the probes had become dirty after eight weeks of use, confirming our study participants' responses.

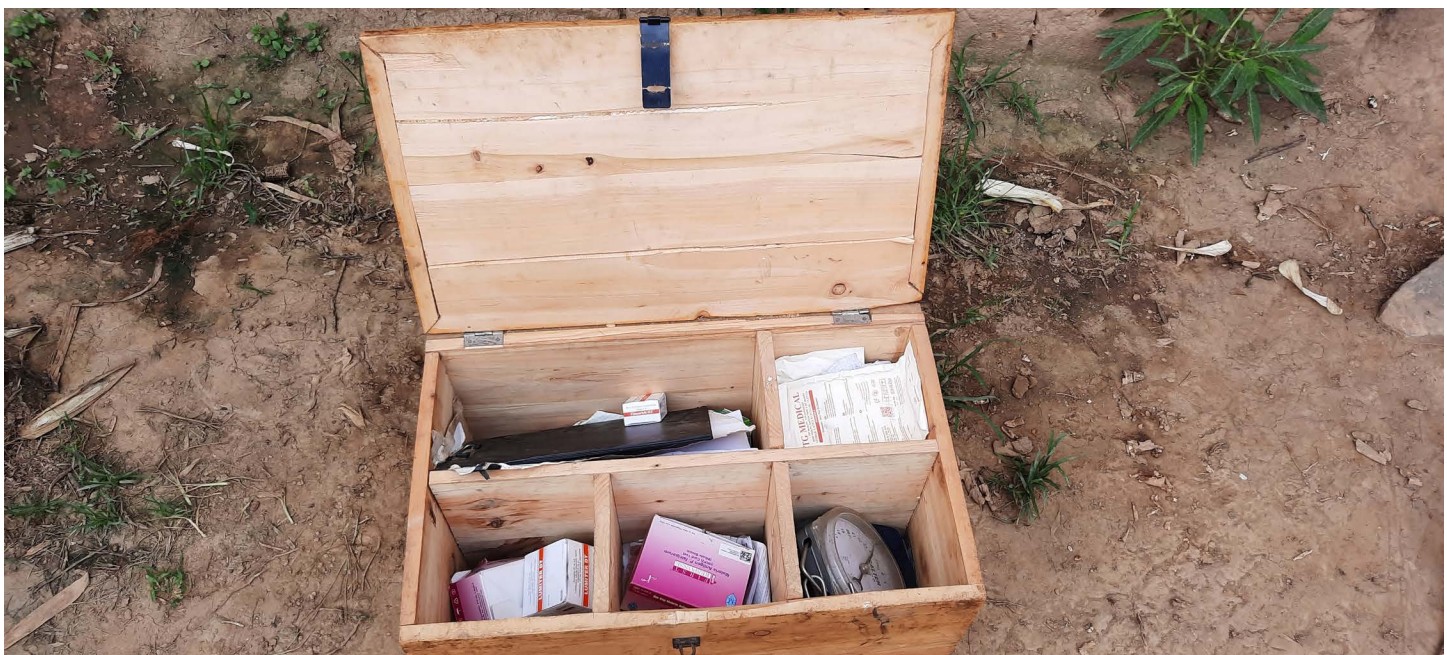

**Fig 2. CHWs stored their devices in metal or wooden trunks provided by the MOH and included medications dispensed during household iCCM consultations.**

### 3.3. Trust in the device

Caregivers and CHWs trusted the parameters collected by the device and felt this improved treatment of childhood illnesses. CHWs also deemed the oximeters more trustworthy than their traditional stopwatches, as the oximeter decreased the risk of manual error, especially when counting respiratory breaths, with one CHW stating:

> 'We do not have to count respiratory movements manually as before as the method brings bias measurement.'

### 3.4. Child agitation and time required to obtain parameters

Most CHWs noted that the average time to obtain parameters using the oximeter was between two to five minutes. However, if a child was agitated, the time could exceed 10 minutes. Child agitation (usually described as crying) was greater among children under one year and a barrier to utilizing the device easily and correctly. Child agitation also prevented quality measurement and led some children to want to remove the probe off their finger, with one participant stating:

> 'Not very easy for the small and turbulent child, but for the child a little old, he could accept that we put the probe on him without problem.'

### 3.5. Desirable pulse oximeter attributes

We also explored the most desirable and appealing attributes of the device among our study participants. Of those FGD participants who responded to this question, the most appealing attribute of the device was the ability to easily and quickly triage or diagnose a child, especially

a child with a fever, as many CHWs lack access to functioning thermometers and stopwatches (Chronos) to support the measurement of respiratory rate. CHWs also noted that their perceived appreciation by the community improved, with one CHW stating:

'It is that we are valued in the community, and now the community sees us as true CHWs.'

Other positive attributes noted by the study participants were perceived improvement in iCCM, accuracy of the device, variety of parameters, and the ability for the device to display results quickly.

### 3.6. Self-efficacy and ability to interpret clinical parameters

While the CHWs were confident in their belief in being able to use the device to collect clinical parameters, they expressed concerns over interpreting the parameters correctly. Many noted that they understood how to interpret and apply temperature as a reference point for diagnosing a sick child, but they lacked understanding on child thresholds for oxygen saturation, respiration rate, and pulse. One CHW stated:

'We are confident (for temperature), but there are questions that cannot be answered without knowing threshold values in order to say this or that another suffers from this disease.'

### 3.7. Sharing the device with other CHWs

We explored the feasibility of CHWs sharing one device, but participants unanimously agreed that sharing the device would be problematic, stating:

'Sharing is not easy because it may affect our work for instance, if we give the device to another CHW and you get a patient at the same time, you will struggle,' and 'It's not easy; we don't live on the same hills and sub-hills. These devices must be numerous, and each must have its own device.'

CHWs also noted that protecting the devices from theft was a personal commitment and expressed fears of work inconveniences, fears of sharing with untrained CHWs, and fears of the borrower mishandling and damaging the device.

### 3.8. CHW and caregiver recommendations for improving the usability of the device

We also asked CHWs and caregivers for recommendations on how to improve the usability of the device. The most common response was to increase the availability of devices followed by building CHW competencies on interpreting clinical parameters during household iCCM consultations. Other recommendations included finding a solution for the slow display of respiratory rate, portable backup battery packs to recharge the devices, and strategies for calming agitated children. Other recommendations included scaling pulse oximeters to health centers, device maintenance plans, solutions for keeping probes clean, a raincoat to shield the device from rain, having a user interface in the local language (Kirundi), and access to a light source (flashlight) to be able to use the device at night.

## 4. Discussion

Participants generally deemed the Rad-G pulse oximeters to be an acceptable, feasible, and valuable tool for ICCM consultations, similar to the findings evaluating barriers and

opportunities of pulse oximetry in Malawi and Bangladesh [27]. Interestingly, our study found CHWs perceived the devices to improve their appreciation and value in the communities they served. Participants also highly preferred the Rad-G pulse oximeter compared to commonly practiced methods of assessing the symptoms of sick children, such as touching the forehead to determine fever and using Chrono watches to count respiratory rate. Our findings suggest that tools that objectively measure pulse oximetry, temperature, and respiratory rate have promise among frontline health workers in rural communities. These tools could also improve iCCM diagnostic practices and inform decision-making around treatment and referral, thereby decreasing the risk of overprescribing antibiotics and facilitating appropriate, timely care.

A commonly cited barrier to using the device was child agitation (expressed as children being afraid of the device or crying), which can prevent accurate measurement of parameters, especially among children six to 12 months. Similar findings around child calming difficulties among health extension workers were reported in a cross-sectional study evaluating usability and acceptability of a multimodal respiratory rate and a pulse oximeter device in case management of children with symptoms of pneumonia in Ethiopia [14]. Infants under one year are more prone to agitation than older age groups due to their limited ability to self-regulate emotions, as their neurobiological systems and coping mechanisms are still developing, making them more sensitive to stress and stimuli [28]. To mitigate these challenges, CHWs who utilized strategies to distract an agitated child, such as games to calm the child, reported greater success in calming the child and collecting the parameters. Thus, including child calming techniques in future frontline health worker pulse oximeter trainings, which could include the use of locally-produced toys to distract children, could be helpful. Devices that play music or have animation could also be an important strategy for calming and distracting a child when collecting parameters.

Issues around safely storing the device were another emerging issue from our research. While CHWs stored their devices in wooden trunks and metal boxes, most preferred keeping them in their original packing boxes. This indicates that the CHWs felt that the metal and wooden trunks did not guarantee safe storage against devices breaking. They often mentioned frustration with the inability of the device box to fit into their trunks, and a few respondents recommended equipping CHWs with backpacks to improve protection and portability of the device. CHWs were also concerned with recharging the device. Most participants complained about the lack of electricity in their communities and that the distance from their homes and the health facilities was a significant barrier to recharging the devices. Several respondents recommended a power bank or a solar source as a solution to recharging their devices. Keeping the oximeter's sensor probe clean was also cited as a maintenance concern and barrier to feasibility in rural household settings. The probes quickly became dirty during use, often due to children's dirty fingers. Cleaning the probes after each use and ensuring the child's fingers are clean before collecting parameters could be potential solutions to this issue.

CHWs' self-efficacy in interpreting the parameters correctly was greatly limited, with CHWs only perceiving to correctly interpret temperature. Another emerging theme from our research was the reluctance to share the device with other CHWs. None of the CHW study participants were willing to share their device with other CHWs within their local area, primarily due to the CHWs' strong sense of responsibility and commitment to owning the device. CHWs also felt that sharing the device with other untrained CHWs might increase the risk of the device being damaged or mishandled. The concern to keep the device safe indicates the high value CHWs had for the Rad-G pulse oximeters and their pride in using a device during iCCM consultations. CHWs recommended increasing availability by acquiring more devices to solve the need to share devices.

### 4.1. Strengths and limitations

A limitation of this study is that it was potentially subject to social-desirability bias, with caregivers and CHWs expressing opinions that they thought the FGD facilitators wanted to hear. To minimize this potential limitation, the purpose of the study was explained to the participants during the consent process, and they were encouraged to share their points of view as truthfully as possible. Another limitation is that we did not separate the CHW FGDs by gender. To mitigate gender bias, FGD facilitators were trained on group dynamics and ground rules to ensure every participant had the opportunity to share in a safe and confidential environment. However, future studies could apply a gender framework when exploring barriers and facilitators to using pulse oximeters during CHW iCCM consultations. An additional limitation was the relatively short follow-up period of eight weeks (from receipt of devices to FGDs) for this pilot. Future studies should evaluate the experiences and perceptions of CHWs and caregivers, as well as device durability and performance, including battery life, over a longer time period, such as 6 to 12 months. However, initial user experiences can impact longer-term adoption and uptake, and our findings provide important insights relevant to implementation considerations. This study was also limited to sick child encounters among children 6–59 months, and additional studies could examine the utility of a multimodal oximeter among children under six months as this age group tends to experience higher rates of severe illness and mortality. Additionally, our study was conducted in a very rural and mountainous part of the country, and our findings may not be generalizable to other settings.

This study has several strengths. There have been limited studies on diagnostic tool use and acceptability among CHWs in low-resource settings. To the best of our knowledge, this is the first study to evaluate the acceptability and feasibility of a multi-diagnostic tool in Burundi. Another strength of the study was our human-centered community-minded focus on CHWs and caregivers, recognizing that caregivers function as the first tier of the healthcare system and, therefore, their experiences and perceptions of new technologies within the continuum of care are critical. The insights of CHW and caregiver pilot participants regarding the oximeter will help inform future device design, training plans, scaling approaches, and integration within the healthcare system. Although designing the device prototype was outside the scope of this study, utilizing human-centred design (HCD) approaches, like the study in South Africa that used HCD to design a smartphone-based pulse oximeter for children [29], can be a valuable method to facilitate pragmatic and tangible design changes.

## 5. Conclusions

Although the CHWs enrolled in the study demonstrated limited knowledge of basic iCCM protocol and lacked simple respiratory rate counters and thermometers which nurses use in local health clinics, they were eager to learn and felt pride in being responsible for their assigned pulse oximeter. Among the communities the CHWs served, the multifunction oximeters increased the perceived value and appreciation of CHWs according to both CHWs and caregivers of local children.

Additional training on iCCM protocols, job aids (sick child decision-making tree), and how to correctly interpret and apply the oximeter parameters could improve the ability of CHWs to diagnose illnesses accurately, resulting in more children receiving appropriate and early treatment for illness and preventing over-prescribing of antibiotics. CHW supportive supervision is also needed to strengthen the community health care system, and there is an urgent need for CHW access to basic diagnostic tools such as thermometers and respiratory rate counters. Finally, because CHWs only conducted 20 to 25 iCCM consultations per week,

the device may be better utilized by nurses at the health-facility level who, in this setting, triage a higher number of children per week (50 to 200). However, more research is needed to determine community-level and health facility-level child morbidity and mortality, care-seeking practices, and where the greatest burden of severe illness is encountered to inform where diagnostic tools such as multifunction oximeters can be best utilized in low-resource settings. Adopting pulse oximetry in sick child consultations within Burundi has the potential to substantially improve the quality of care by enabling early detection of hypoxemia, allowing for timely interventions and better management of severe respiratory conditions. Future studies, utilizing HCD approaches, should evaluate challenges and opportunities to bringing pulse oximetry to scale at community and facility levels, and consider issues such as device design, durability, adaptive training and support strategies, cost-effectiveness, and performance over a longer time period.

## Supporting information

**S1 File. Caregiver FGD guide.**
(DOC)

**S2 File. CHW FGD guide.**
(DOC)

**S3 File. Codebook.**
(DOCX)

**S1 Checklist. Inclusivity in global research.**
(DOCX)

## Acknowledgments

We are thankful for the support of Dr. Olivier Nijimbere, Burundi MOH Permanent Secretary, Dr. Clement Djumo and Dr. Nestor Nkengurutse, Burundi UNICEF, and Dr. Joe Gallo, Johns Hopkins University Bloomberg School of Public Health.

## Author contributions

**Conceptualization:** Sarah Bauler, Dionis Nizigiyimana, Asrat Tolossa.

**Data curation:** Sarah Bauler, Leocadie Nivyindika, Vital Habonimana.

**Formal analysis:** Sarah Bauler, Titus Kirwa.

**Funding acquisition:** Sarah Bauler, Asrat Tolossa.

**Investigation:** Sarah Bauler, Leocadie Nivyindika, Vital Habonimana, Asrat Tolossa.

**Methodology:** Sarah Bauler, Dionis Nizigiyimana, Asrat Tolossa.

**Project administration:** Sarah Bauler, Leocadie Nivyindika, Asrat Tolossa.

**Resources:** Sarah Bauler.

**Software:** Sarah Bauler, Titus Kirwa.

**Supervision:** Sarah Bauler, Leocadie Nivyindika, Vital Habonimana.

**Validation:** Sarah Bauler, Asrat Tolossa.

**Writing – original draft:** Sarah Bauler, Miles A. Kirby.

**Writing – review & editing:** Sarah Bauler, Titus Kirwa, Vital Habonimana, Dionis Nizigiyimana, Miles A. Kirby, Asrat Tolossa.

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
