## [Decision Letter · Decision Letter 0]

5 Jun 2024

PGPH-D-23-01703

Community health worker and caregiver experiences and perceptions of a multimodal handheld pulse oximeter used in sick child consultations in rural Burundi: A qualitative evaluation

Dear Dr. Bauler,

Thank you for submitting your manuscript to PLOS Global Public Health. After careful consideration, we feel that it has merit but does not fully meet PLOS Global Public Health’s publication criteria as it currently stands. Therefore, we invite you to submit a revised version of the manuscript that addresses the points raised during the review process.

One reviewer has provided their feedback on your manuscript, please consider their suggestions and make the necessary amendments. 

Please note that we have only been able to secure a single reviewer to assess your manuscript. We are issuing a decision on your manuscript at this point to prevent further delays in the evaluation of your manuscript. Please be aware that the editor who handles your revised manuscript might find it necessary to invite additional reviewers to assess this work once the revised manuscript is submitted. However, we will aim to proceed on the basis of this single review if possible. 

We look forward to receiving your revised manuscript.

Kind regards,

Joanna Tindall

Staff Editor

Journal Requirements:

2. In the ethics statement in the Methods, you have specified that verbal consent was obtained. Please provide additional details regarding how this consent was documented and witnessed, and state whether this was approved by the IRB. 

3. Please include a complete copy of PLOS’ questionnaire on inclusivity in global research in your revised manuscript. Our policy for research in this area aims to improve transparency in the reporting of research performed outside of researchers’ own country or community. The policy applies to researchers who have travelled to a different country to conduct research, research with Indigenous populations or their lands, and research on cultural artefacts. The questionnaire can also be requested at the journal’s discretion for any other submissions, even if these conditions are not met.  Please find more information on the policy and a link to download a blank copy of the questionnaire here: https://journals.plos.org/globalpublichealth/s/best-practices-in-research-reporting. Please upload a completed version of your questionnaire as Supporting Information when you resubmit your manuscript.

4. Please send a completed 'Competing Interests' statement, including any COIs declared by your co-authors. If you have no competing interests to declare, please state "The authors have declared that no competing interests exist". Otherwise please declare all competing interests beginning with the statement "I have read the journal's policy and the authors of this manuscript have the following competing interests:"

5. Please ensure that Funding Information and Financial Disclosure Statement are matched.

6. In the Funding Information you indicated that no funding was received. Please revise the Funding Information field to reflect funding received.

7. Please provide separate figure files in .tif or .eps format only and remove any figures embedded in your manuscript file. Please also ensure all files are under our size limit of 10MB.

8. We have noticed that you have uploaded Supporting Information files, but you have not included a list of legends. Please add a full list of legends for your Supporting Information files after the references list.

9. In the online submission form, you indicated that "Data and research instruments available upon request". 

3. Uploaded as supplementary information.

Additional Editor Comments (if provided):

Reviewers' comments:

Reviewer's Responses to Questions

**Comments to the Author**

1. Does this manuscript meet PLOS Global Public Health’s publication criteria?

Reviewer #1: Yes

2. Has the statistical analysis been performed appropriately and rigorously?

Reviewer #1: N/A

3. Have the authors made all data underlying the findings in their manuscript fully available (please refer to the Data Availability Statement at the start of the manuscript PDF file)?

Reviewer #1: Yes

4. Is the manuscript presented in an intelligible fashion and written in standard English?

Reviewer #1: Yes

Reviewer #1: Thanks for the opportunity to review this important piece. Overall very well written and presented study. A few comments to think about and respond to:

- you had the CHWs use the tool for 8 weeks before the FGDs - was this enough time for them to really understand and use it enough to provide rich and complete data? If you have a rationale for this it would be good to state it. It feels a bit of a short time to me and could almost be a limitation - worth considering and addressing in the article.

For the FGDs I couldn't see if you split them between men and women as sometimes there can be a gender bias presented by mixed groups?

BW

Kevin

**Do you want your identity to be public for this peer review?** For information about this choice, including consent withdrawal, please see our Privacy Policy

Reviewer #1: No

---

## [Decision Letter · Decision Letter 1]

11 Sep 2024

PGPH-D-23-01703R1

Community health worker and caregiver experiences and perceptions of a multimodal handheld pulse oximeter used in sick child consultations in rural Burundi: A qualitative evaluation

Dear Dr. Bauler,

Thank you for submitting your manuscript to PLOS Global Public Health. After careful consideration, we feel that it has merit but does not fully meet PLOS Global Public Health’s publication criteria as it currently stands. Therefore, we invite you to submit a revised version of the manuscript that addresses the points raised during the review process.

Please address reviewer 2's comments for revisions to improve the reporting and clarify the aims of your study.

We look forward to receiving your revised manuscript.

Kind regards,

Jennifer Tucker, PhD

Staff Editor

Additional Editor Comments (if provided):

Reviewers' comments:

Reviewer's Responses to Questions

**Comments to the Author**

Reviewer #1: All comments have been addressed

Reviewer #2: (No Response)

publication criteria?

Reviewer #1: Yes

Reviewer #2: Yes

3. Has the statistical analysis been performed appropriately and rigorously?

Reviewer #1: N/A

Reviewer #2: I don't know

4. Have the authors made all data underlying the findings in their manuscript fully available (please refer to the Data Availability Statement at the start of the manuscript PDF file)?

Reviewer #1: Yes

Reviewer #2: No

5. Is the manuscript presented in an intelligible fashion and written in standard English?

Reviewer #1: Yes

Reviewer #2: Yes

Reviewer #1: Thanks for addressing my comments.

BW

Kevin

Reviewer #2: The paper investigates the acceptability and feasibility of using a multimodal handheld pulse oximeter by Community Health Workers (CHWs) in rural Burundi during sick child consultations. The study focuses on CHWs' and caregivers' experiences using the device to assess children aged 6-59 months as part of Integrated Community Case Management (iCCM). The oximeter measures multiple parameters including pulse rate, oxygen saturation, respiration rate, and temperature, aimed at improving the diagnosis of childhood illnesses, particularly pneumonia. It is a well-written manuscript. However, the current version needs to be revised before it is considered for publication.

1. The paper should define more precise research questions or hypotheses upfront. While the study explores various aspects of pulse oximeter use, a clearer articulation of objectives will help frame the findings more effectively.

2. Kindly expand on the rationale for choosing the sample size of 32 CHWs and clarify whether this size is sufficient for saturation in qualitative analysis. Including more information on how sampling choices may have influenced the findings would be beneficial.

3. The paper notes a particular issue with younger children (6-12 months) but lacks explanation as to why this group was especially problematic. More in-depth analysis of age-specific challenges could help strengthen the discussion.

4. The paper makes comparisons with other studies (e.g., on respiration rate accuracy in other countries). More direct comparisons between the study's findings and those in similar contexts could enrich the analysis and provide a broader perspective.

6. While the paper mentions that pulse oximetry may be more feasible at the health facility level, this argument could be expanded. A more detailed exploration of the challenges to scaling the use of oximeters at the community level, as well as possible solutions, would add depth to the discussion.

8. The paper would benefit from a more thorough discussion of its limitations. For example, limitations in terms of sample size, generalizability to other regions, or potential biases introduced by the qualitative methodology should be explicitly addressed.

9. There are issues with the referencing (e.g., inaccurate references regarding pneumonia burden). Ensuring all references are accurate and align with the claims made in the paper will increase its credibility.

10. The conclusion should more clearly synthesize the implications of the findings. A stronger link between the findings, recommendations for policy/practice, and suggestions for future research would provide a more impactful ending.

**Do you want your identity to be public for this peer review?** For information about this choice, including consent withdrawal, please see our Privacy Policy

Reviewer #1: **Yes: ** Kevin Baker

Reviewer #2: **Yes: ** Yasir Bin Nisar

---

## [Editor Report · Decision Letter 2]

26 Nov 2024

Community health worker and caregiver experiences and perceptions of a multimodal handheld pulse oximeter used in sick child consultations in rural Burundi: A qualitative evaluation

PGPH-D-23-01703R2

Dear Mrs. Bauler,

We are pleased to inform you that your manuscript 'Community health worker and caregiver experiences and perceptions of a multimodal handheld pulse oximeter used in sick child consultations in rural Burundi: A qualitative evaluation' has been provisionally accepted for publication in PLOS Global Public Health.

Best regards,

Bipin Adhikari, MBBS, DTM&H, MCTM, MPH, DPhil

Academic Editor